# Effects of Physical Exercise on the Stereotyped Behavior of Children with Autism Spectrum Disorders

**DOI:** 10.3390/medicina55100685

**Published:** 2019-10-14

**Authors:** José Pedro Ferreira, Thaysa Ghiarone, Cyro Rego Cabral Júnior, Guilherme Eustáquio Furtado, Humberto Moreira Carvalho, Aristides M. Machado-Rodrigues, Chrystiane Vasconcelos Andrade Toscano

**Affiliations:** 1Sport and Physical Activity Research Center (CIDAF), University of Coimbra, 3040-156 Coimbra, Portugal; furts2001@yahoo.com.br (G.E.F.); chrystoscano@gmail.com (C.V.A.T.); 2Statistic Department, Federal University of Alagoas, Maceió CEP 57072-970, Brazil; thaysa_tg@hotmail.com; 3Health Sciences Department, Federal University of Pernambuco, Recife CEP 50670-901, Brazil; cyrorcjr@gmail.com; 4Faculty of Physical Education, Federal University of Santa Catarina, Florianópolis CEP 88040-900, Brazil; hmoreiracarvalho@gmail.com; 5High School of Education, Polytechnic Institute of Viseu, 3504-501 Viseu, Portugal; a.machadorodrigues@esev.ipv.pt; 6Department of Physical Education, Federal University of Alagoas, Maceió CEP 57072-970, Brazil

**Keywords:** meta-analysis, physical exercise, intervention, stereotypes, autism

## Abstract

*Background and Objectives*: Recent studies have shown the existence of a positive relationship between physical exercise, symptomatic improvement, and reduction of damage caused by comorbidities associated with autistic spectrum disorder (ASD) in children, adolescents, and adults. The aim of this systematic review with meta-analysis (SRM) was to estimate the effects of physical exercise (PE) on the stereotyped behaviors of children with a diagnosis of ASD in intervention studies. *Materials and Methods*: The design followed the PRISMA guidelines and the TREND statement to assess the quality of information in each study. Nine non-randomized intervention trial studies with low, moderate, and vigorous physical exercise, with a duration varying from 8 to 48 weeks and a frequency of 3 times a week, were included in the SRM. The dependent variable episodes of stereotypical behaviors was analyzed in all studies and assessed as the number of episodes demonstrated by the child in pre- versus post-exercise intervention conditions. *Results*: The eight studies included a total 129 children (115 males and 14 females) with an average age of 8.93 ± 1.69 years. Children with ASD showed a reduction of 1.1 in the number of occurrences of stereotypical behaviors after intervention with physical exercise. *Conclusion*: Evidence was found to support physical exercise as an effective tool in reducing the number of episodes of stereotypical behaviors in children diagnosed with ASD.

## 1. Introduction

The term autistic spectrum disorder (ASD) refers to a complex category of the neurobiological development disorders typically diagnosed during childhood [1]. It also includes deficits in many aspects of social reciprocity, pragmatic communication deficits and language delays, and an assortment of behavioral problems, such as restricted interests, sensory sensitivities, and repetitive behaviors [2]. The causes include genetic events, metabolic disorders, infectious diseases, neuroanatomical and biochemical structural abnormalities in the brain as well as others still being researched [3].

The estimated prevalence of ADS is 1% in the North American population [4] and 0.3% in the Brazilian population [5]. Estimates suggest that ASD occurs more commonly in males than females, with a gender ratio of 4.3:1 across the full intelligence quotient (IQ) range [6]. Additionally, the prevalence rates may differ between countries specially in the case of under development countries where the lack of specialized services for the early detection of ASD contribute for lower prevalence rates [4,5]. The intensity of the ASD symptoms can range from mild to very severe [7].

Several studies have indicated different treatment techniques used to minimize the effects and the intensity of the symptoms [8,9,10], including antecedent-based treatments, consequence-based treatments, extinction-based treatments, and combinations of treatment strategies [11], psychopharmalogical-based treatment [12], and the use of physical exercise [13].

The most common stereotypical behaviors are rocking motion of the hands, nodding or shaking arms, sudden runs, body balance forward and backward, repeated manipulation of objects and finger movements [14], standing out from their significant interference in social interactions and learning during childhood [14,15,16,17]. These movements are involuntary, with an exclusive function of producing physical and sensorial self-regulation, limiting individual’s interaction with the environment [18]. Such behavior look unusual or strange to the average person, but are characteristic of the individuals with ASD and do not cause any physical harm (e.g., rocking, spinning, and manipulating objects repeatedly) [14].

Treatment options directed to the reduction of stereotypies, in individuals with ASD, are often highly intrusive. Psychotropic medication [19] and intensive behavioral interventions [20] are the most common treatments. The impact of the treatments in the reduction of stereotypical behavior is assessed based on the number of stereotypical episodes occurring over a period of time, i.e., the number of times repetitive voluntary acts are outcast during a limited observation period of time. 

In the 1970s, evidence for the positive effects of physical exercise (PE) on stereotyped behavior of children with ASD was reported for the first time [21]. Several studies have strengthened the case for the use of physical exercise, a subset of physical activity that is planned, structured, and repetitive, and done with the purpose of improving or maintaining physical fitness, as an optional or adjunct treatment [15,16,22,23,24,25,26]. Several systematic reviews [27,28,29,30] and meta-analyses [31,32] in the literature supported the idea of the positive effects of physical exercise in groups composed exclusively of children [29], children and adolescents [28,31], and with a greater age range including children, adolescents, and adults [27,32]. Additionally, there was also a dissemination of specific recommendations for the development of physical exercise programs for people with ASD [30].

Bremer and colleagues [27] highlighted the lack of evidence of the effects of exercise in the early childhood. Specifically, they recommend further future intervention research with more controlled methodological designs, standardized assessments, larger samples, and longitudinal follow-ups in order to better understand the effects of different types of physical exercise and the extent of its benefits in children with ASD.

Different studies showed positive effects of physical exercise in different symptomatology and comorbidity categories. Such studies have identified: (a) reductions in physical motor deficits [33,34] and obesity and overweight [35,36,37]; (b) psychopathological improvements with an increasing time of concentration in the execution of tasks at the table [38] and improvements in cognitive function [27,32]; (c) behavioral improvements with stereotypies [16,23] and reductions in aggressive behaviors [26,27]; and (d) improvements in socio-emotional function [27].

Moreover, effects of physical exercise on stereotypic behavior of children with ASD have always been shown to sharply decrease immediately after physical exercise-based interventions [15,23,39], and gradually increase and return to baseline levels over a period of rest. However, the length of such time effects is not consensual in the literature varying from 40 min [24,40] to 90 min [16] and to 120 min [41].

The purpose of this systematic review with meta-analysis is to identify the magnitude of the effects of the intervention programs with physical exercise in stereotypical behaviors of children with a single diagnosis of autism spectrum disorder (ASD). In the present study, the term physical exercise was defined as a subset of physical activity that is planned, structured, and repetitive; has a final or an intermediate objective; and involves the improvement or maintenance of physical fitness [42] through the participation in different regular physical activities (e.g., running, swimming, rowing, cycling, ballroom dancing, or walking faster than 5 km/h).

## 2. Methods

### 2.1. Search Strategies

Search strategies followed the PRISMA guidelines [43] and were based on the following descriptor terms and keywords defined by the authors and indexed to the Medical Subject Headings (MESH, U.S. National Library of Medicine, 8600 Rockville Pike, Bethesda, MD 20894): “autism spectrum disorder,” “autistic,” and “Asperger’s syndrome,” combined with terms “exercise,” “physical fitness,” and “aerobic exercise.” Combinations of these keywords were inserted into searches of the following academic journal data bases: US National Library of Medicine National Institutes of Health (U.S. National Library of Medicine, Rockville Pike, Bethesda, MD 20894; https://www.ncbi.nlm.nih.gov/pubmed/), Education Resources Information Center (Institute of Education Sciences at the U.S. Department of Education, Washington, DC 20024; https://eric.ed.gov/), PsycINFO (American Psychological Association, Washington, DC 20002-4242; https://www.ebsco.com/products/research-databases/psycinfo), and Web of Science (Clarivate Analytics, Boston, MA 02210; http://apps.webofknowledge.com). The advanced meta-search option was carried out using the resources inherent to each database. The research procedures were carried out between 1st of March and 31st of December 2017. 

### 2.2. Data Extraction

Two different researchers conducted the initial search using a list of keywords. The following screening procedures were implemented to determine whether the articles from the initial search were relevant for the study: (a) reading the titles, and if titles seemed relevant then citations were saved in specific software (Mendeley Desktop v.1.19.04, London, UK). After the initial screening, all duplicates were removed; (b) reading the abstracts, and if they did not provide enough information about the topic, then articles were excluded from the study; (c) reading articles in full-text and if they did not meet the inclusion criteria then were excluded from the study; (d) in case of an article classified as “doubtful,” i.e., one of the researchers decides to include it in the study while a second researcher decides to exclude it from the study, a third independent reviewer with expertise in systematic reviews was asked to analyze the article, to discuss the application of the inclusion/exclusion criteria, and to obtain consensus toward its inclusion or exclusion in the systematic review with meta-analysis; and (e) checking the quality of the information (QoI) from each study using the TREND statement [44,45,46] as a final eligibility criteria.

If TREND items and sub-items obtained a total score equivalent to a total percentage below 50%, manuscripts were excluded from the study due to lack of quality of information. Additionally, if a full-text article provided incomplete data, authors were contacted by e-mail requesting missing information. If no response was obtained, the article was excluded from the study.

### 2.3. Criteria for Study Selection

Several inclusion criteria were used to determine which articles should be selected for the present study during initial screening. The inclusion criteria for studies were: (a) samples composed exclusively of children and youngsters with a diagnostic of autistic spectrum disorders (Asperger’s syndrome, autism or unspecified developmental disorder) with an age group up to 16 years of age; (b) using physical exercise as an intervention tool and without any other parallel secondary type of intervention (e.g., cognitive or social intervention) or without the participation of animals as a complementary therapeutic resource; (c) used stereotypical or self-stimulatory behavior as a dependent variable; and (d) published between the years 1970 to 2017.

Due to the restrictive nature of the inclusion criteria and the specificity of the topic under research, i.e., the magnitude of the effects of the intervention programs with physical exercise in the stereotyped behavior of children with a single diagnosis of autistic spectrum disorder (ASD), it was expected that relatively few studies in this area would likely be found.

### 2.4. Methodological Design

The PRISMA Statement [47] positioning guidelines were followed to assist the design of this study. These guidelines describe the four stages (identification, screening, eligibility, and final selection) required to search and select manuscripts for a systematic review and feature the option of illustrating procedures in a flowchart [47]. Qualitative data from the different articles was selected, extracted, and organized in a specific table, following the PRISMA method including author´s year and country, number of participants included, their age and sex, ASD diagnosis, the type of intervention study, central outcomes, and the existence of a control group.

### 2.5. Quality of Information (QoI)

Moreover, an evaluation of the quality of the information (QoI) from the articles included in the systematic review based on the application of the TREND positioning guidelines (Transparent Evaluation Report with Nonrandomized Designs) [44,45] was carried out. The method requires evaluation of a list of 22 items (general criteria) subdivided into 59 sub-items (specific criteria) able to quantitatively assess the QoI [48]. One point is given to each item and sub-item completed. The checklist was conducted by two researchers separately. A minimum criterion of a QoI ≥ 50% was established to allow the article to be included in the final meta-analysis, qualifying it as an article of high relevance for the topic under study.

### 2.6. Publication Bias

The publication bias for the final set of studies in the present systematic review with meta-analysis (SRM) was calculated in Comprehensive Meta-Analysis software creating a funnel plot by the standard error (y-axis) and the standard difference in means (x-axis) to determine whether the plot was balanced. Funnel plots are either symmetrical or asymmetrical [49,50]. Studies without publication bias are distributed symmetrically around the mean effect size, since the sampling error is random. Studies with publication bias are expected to follow the model with symmetry at the top of the funnel plot, a few studies missing in the middle, and more studies missing near the bottom of the plot. If the direction of the effect is toward the right, then near the bottom of the funnel plot we expect a gap on the left, where the non-significant studies would have been if we had been able to locate them. Because the interpretation of the funnel plot is sometimes subjective, different tests such as the Begg, the Mazumdar, and the Egger have been proposed to quantify bias and test the relationship between sample size and effect size [51,52]. In the present study, the Egger´s test was used to check publication bias as suggested by Borenstein et al. [53].

### 2.7. Effect-Size Calculations

Effect size was calculated using the software Comprehensive Meta-Analysis (CMA) (Biostat, Englewood, NJ, USA, version 3.3.070), November 2014. The effect-size metric selected was the standardized difference in means (standard difference in means) as all studies evaluated the same outcome variable, but with different criteria. In such circumstances, it is necessary to standardize the results from each study using a uniform scale before they can be combined [54].

Data extracted for effect-size calculations from the different studies included pre- and post-means (M), standard deviations (SD), sample size (N), and effect direction. If these data were not available, F-values and pre–post correlation values were extracted and used. A random-effects model was used for the present meta-analysis as it combines sampling error and between-study variance to estimate effect size [54]. When means, standard deviations, and sample size values were reported in the manuscript, effect sizes (Cohen’s d) were calculated for the central outcome of studies (occurrence of stereotyped behavior). The following thresholds were used to interpret the effect sizes: trivial (d ≤ 0.20), small (0.21 < d < 0.50), moderate (0.51 < d < 0.79), and large (d > 0.80) [53].

### 2.8. Heterogeneity of Variance

For this SRM, we followed the assumption that there would be variability in the true effect sizes between studies due to the expected differences in sampling error and between-study variance. The following statistics were used to quantify between-study heterogeneity: Q-value, I-squared (I^2^), tau-squared (τ^2^), and tau (τ). The Q Cochran statistic was used as a significance test to verify the null hypothesis and assess if all manuscripts involved in this SRM share common effect sizes. Any variation would be due to the sample error within the studies. If all studies share the same effect size, the expected Q value will be equal to the degrees of freedom (df), i.e., the number of studies minus 1. The I^2^ statistic corresponds to the ratio of the true heterogeneity of the total variation of the observed effects. It shows the proportion (percentage) of the observed variance that reflects the differences in the true effect size rather than in the sample error [55]. The τ^2^ is the variance of the true effect sizes (in log units) among studies, while the τ value refers to the standard deviation of the true effects [54].

## 3. Results

### 3.1. Study Selection

Figure 1 shows the flowchart representing the four stages of the systematic search using the PRISMA statement guidelines (see Figure 1).

In the identification stage, 310 articles were selected. In the screening stage and after reading the titles and abstracts, 171 articles were excluded because they were replicated, 99 articles were excluded after initial screening, including title and summary reading due to different reasons—90 articles had keywords which were unrelated, featured the wrong age group and/or diagnostic variability, 4 articles were case studies, and 5 articles had included the participation of animals as a therapeutic resource (dogs and horses). In the eligibility stage, 40 studies were analyzed and after reading the full articles, 7 articles were review articles, including 1 ahead-of-print meta-analysis, 15 articles were not intervention studies or were not assessing stereotypies as a behavioral outcome, 1 article was an ahead-of-print study protocol, and 9 articles were excluded because they used children with other behavior conditions together with children with ASD in the same intervention group making impossible to separate the data. The TREND methodology guidelines were applied to the five eligible articles included in the final revision to assess the QoI in this review. Results higher than 50% of QoI were accepted and regarded as satisfactory for the quantitative evaluation and included in the SRM (see Table 1).

### 3.2. Characteristics of the Participants

The total sample analyzed included 129 children diagnosed with autism. The average age was 8.93 ± 1.69 years. Studies have selected the participants (see Table 2) using various criteria which included: (a) completed autism diagnosis based on medical expertized assessment [14,15,40]; (b) children with high stereotyped behavior [14,15,40]; (c) capacity to be submitted to intelligence tests, such as Vineland [14]; and (d) good health and physical capacity attested by a doctor [14,40]. Three studies reported carrying out an intervention with physical exercise (e.g., jogging, jumping on a trampoline, stationary biking, completing an obstacles course, playing dance or revolution game, participating in a basic coordination, and strength exercise program) in a school environment [15,40]. Others studies reported the use of physical exercise (e.g., ball exercises and treadmill or stationary biking) in a therapeutic environment [56] and controlled conditions [57].

### 3.3. Meta-Analysis Outcomes

The eight studies (nine entries) included in this SRM were all intervention studies where ASD participants were exposed to physical exercise. One of the studies [16] presented separate results from two sub-samples of male and female individuals. Both were used in the meta-analysis. In spite of fulfilling all the criteria defined for the present study, data from the study by Neely et al. [25] was not included in the present study as the results were presented in a format not suitable to be used in the meta-analysis.

All eight studies followed the same research hypothesis testing whether intervention programs with physical exercise would produce a positive effect on the stereotyped behavior of children with autistic spectrum disorder, confirming a hypothetical reduction on the frequency of stereotyped behaviors’ episodes. The variation of the results was analyzed based on the average number of stereotyped behavior episodes checked before and after the intervention with physical exercise. The effect size is represented here by the standard difference in means, expressing the effect size of the intervention in each study regarding the variability observed in the same study. In fact, the intervention effect is the result of the difference in means and not the result of the mean of the differences.

### 3.4. Magnitude of the Difference Between Pre- and Post-Means

This systematic review with meta-analysis showed, based on the standard difference in means, that children with ASD showed a reduction that equals to 1.110 in the number of occurrences of stereotypical behaviors before and after intervention with physical exercise (see Figure 2), when the inclusion and exclusion criteria previously described were met.

The confidence interval for the standard difference in means is 0.277 to 1.943, meaning that the raw mean difference, may fall anywhere in this range. Moreover, this interval does not include zero difference. Similarly, z-values obtained to test the null hypothesis, that the standard difference in means is zero, showed a z = 2.611, and a corresponding value of *p* = 0.009 (*p* < 0.05). Thus, the null hypothesis was rejected and the alternative hypothesis accepted in all analyzed studies, and after interventions with physical exercise, there is a reduction in the incidence of stereotypical behaviors in children with autistic spectrum disorder, with a standard difference in means before and after the intervention higher than 1 point.

### 3.5. Homogeneity of the Effects

Another relevant question for this study is to assess if the effect size varies across studies. It is known that the observed effect size varies somewhat from study to study and that a certain amount of variation is expected due to sampling error. Therefore, it is necessary to determine whether the observed variation falls within the range that can be attributed to sampling error, which would result in a lack of evidence for the variation in true effects, or if it exceeds that range.

The Q-value statistic was used to test the null hypothesis that all studies included in this SRM shared a common effect size, and if any variation would be due to the sample error within the studies. If all studies shared the same effect size, the expected value of Q would be equal to the degrees of freedom (df). The obtained Q value is 21,741 with 8 degrees of freedom, and *p* = 0.005 (*p* < 0.05). Thus, the null hypothesis must be rejected as the true effect-size is not identical in all studies. While the observed variation falls within the range that could be due to the sampling error, our estimate of the variance in true effects may be zero, as presented in the following statistics.

The I^2^ value, representing the ratio of the true heterogeneity of the total variation of the observed effects, is 63.210 meaning that about 63.21% of the variance in the observed effects reflects variance in the true effects. The τ^2^ value, representing the variance of the true effect-sizes (in log units) among studies, has a value of 0.839. The τ value, i.e., the standard deviation of the true effects in this SRM is equal to 0.916.

Multiple methods exist to detect the presence of publication bias and to assess its impact on the different studies from this SRM. The visual observation of the funnel plot (Figure 3) was used, representing the effect size of each study in relation to its standard error.

The funnel plot for the distribution of the observed studies is noticeably symmetric, with the majority of the studies distributed symmetrically about the mean effect size, since the sampling error is random, providing no subjective evidence of publication bias. The distribution of studies appears symmetrical in relation to the effect size of the true population, and the plot becomes narrower as the standard error decreases. Additionally, the Egger´s test was used to test the null hypothesis according to which the intercept is equal to zero in the population. For the present SRM, the intercept is 1.51537 with a 95% confidence interval between −0.78035 and 3.81110, a t-value = 1.56085 and df = 7. The recommended *p*-value (2-tailed) is 0.16253. Thus, there is also no statistical evidence for publication bias.

## 4. Discussion

The present systematic review and meta-analysis (SRM) aimed at the evidence from existing studies for the effect of physical exercise on the occurrence of stereotypical behavior in children with ASD. The outcome adopted in the nine studies was assessed as the difference in the number of stereotyped or self-stimulatory episodes at baseline (the child is observed without any interference from the teacher for the counting of stereotyped behavior) and after intervention. Physical exercise provided the intervention stimulus.

In the studies included in this SRM, it is possible to identify the presence of only 14 girls [7,15,37,41]. The low participation of females in the intervention groups can be explained by the higher incidence of ASD in boys, i.e., 4 to 8 male subjects for each female [58].

From the nine entries analyzed in this SRM, eight showed the existence of a positive relationship between physical exercise and the reduction of stereotypical behaviors in children with ASD. Only one of the eight studies did not find a reduction in stereotypic behaviors after physical exercise [38]. The higher the repertory of stereotypical behaviors in children with ASD, the lower the chances to understand and explore the environment and consequently develop new learning skills [15,25,34]. Some studies in the literature revealed that about 44% of children with ASD have at least one type of stereotypy [13]. In the eight studies analyzed (nine entries), seven identified the presence of stereotypies in 100% of children in the sampling group [14,15,16] and one [37] found that only 55.6% of the total sample had stereotypies. In this case, data from the stereotypies sub-group was used for the present analysis. The concept of stereotypies adopted in the studies included in this SRM is based on the same theoretical background, that is, the stereotypical behaviors are reflex responses, without motor function in a limited capacity of social effect [14,15,16,37,40]. All the studies described the stereotypies profile of the sampling group [14,15,16], some subdividing them into motor and vocal/oral categories [15], and others into repetitive and self-stimulatory, vocal responses, and self-injuries [14,15,16,37].

From the studies identified in this SRM, eight (nine entries) confirmed the hypothesis that the specific types of exercises when articulated to specific types of stereotypical movements had better effects in reducing stereotypies [14,15,16], and two studies (three entries) confirmed the hypothesis that vigorous intensity physical exercise cause greater reductions when compared to moderate intensity physical exercise [14,15]. However, one study [57] revealed that aerobic exercise performed at low intensity (50–60% of the age-calculated maximal heart rate) generated a better and more consistent reduction of repetitive and restrictive behaviors, also known as stereotypical or self-stimulatory behaviors (SSB), at baseline. Only two of the eight studies (nine entries) classified physical exercise as aerobic, highlighting the importance of monitoring the exercise effort performed by the children during the intervention [37].

Regarding the characteristics of the physical exercise, studies included games [14,16], games and walking [15], games and running [37], activity stations such as jumping, trampoline, dancing [40] or strength, balance, and coordination exercises [40], stationary biking [41], and martial arts techniques [7]. The intensity of the exercise was defined as low or high in one study [58] and moderate or vigorous in three of the eight analyzed studies [14,15,41]. The interventions had an average frequency of three times per week, with the duration per session that could vary from 15 to 90 min per session and exercise programs varying from 8 to 48 weeks. The PE was used, in these eight studies (nine entries), as an antecedent of the activity whose performance should be improved based on the reduction of stereotypical or self-stimulatory behaviors.

The present meta-analysis also showed that due to the relative weight of some of the studies included [7,41,57] they should be discussed in further detail as they provide new relevant and complementary information to explain the role and the effects of exercise in the reduction of the number of episodes of stereotypic behavior in children with ASD. The purpose of study by Liu et al. [41] was to determine the effects of physical activity engagement on stereotypic behavior in children with ASD and to quantify the length of time the effect would last. A moderate to vigorous physical activity (MVPA) exercise program was used as intervention involving children with ASD in activities such as run, bike ride, or walk in the park for 15 min. Results showed a significant decrease on the stereotypic behavior of children with ASD which may last for at least two hours. The calculated effect-size was large (d = 1.35) indicating that the true effects in the population might be considerable. These findings were consistent with previous studies reporting exercise to be beneficial to children with ASD stereotypic behavior [7,14,15].

A study by Bahrami et al. [7] also tested the effects of an exercise program (Kata techniques) in children with ASD and concluded that it led to consistent reduction of their stereotypic behaviors in 42.54% of participants, in the exercise group. This specific exercise program was based on a modified form of Heian Shodan Kata (Shotokan) used as experimental task. The Kata of karate (meaning ‘form’) is a sequence of techniques specific from each school of martial arts integrating logical arrangements of blocking, punching, sticking, and kicking techniques in a set sequence, i.e., a specific number of movements performed in a particular order, with participants moving in several directions in space [7]. There is some variation in the complexity and the time required to complete the movements, each one with its own meaning and function. While performing a Kata, the karateka should imagine self to be surrounded by opponents and be prepared to execute defensive and offensive techniques in any direction [59]. To execute this Kata, participants are required to be familiar with concepts of individual hand and foot motions, basic stances, basic kicking techniques, and the processes involved in advancing, retreating, and turning the body about in a stable fashion [60]. The calculated effect size was large (d = 0.844) indicating that the true effects in the population might have moderate implications. Additionally, this study provided further evidence to understand the effect of exercise duration in children with ASD showing that after 30 days of no practice, stereotypy in the exercise group remained significantly lower than in pre-intervention time.

A study by Nazemzadegan et al. [57] indicated that the intervention with ball training (Jim ball) resulted in changes in the stereotypical behavior of children with high functioning autism spectrum disorder. Thus, the general study results indicated the effectiveness of ball training exercises on the stereotyped behavior of the participants.

Such results put into question the assumption that stereotypic behaviors will sharply decrease and eventually return to baseline levels after physical-exercise-based interventions are completed [14,22,39]. The length of such time effects is not consensual in the literature varying around 40 min [23,61], 90 min [15], or 120 min [41]. One of the major contributions of the study by Bahrami et al. [7] is to provide new evidence that some types of exercise such as Kata techniques, due to its specific characteristics, may have an effect for a longer duration on the reduction of stereotypic behavior in children with ASD. In this specific case, the effect lasted at least for 30 days.

Previous researchers often associated the decrease of stereotypic behavior to the effects of fatigue resulting from exercise; however, in the study by Bahrami et al. fatigue did not seem to be a possible explanation for the assessment of stereotypies two days after the last exercise session and again after thirty days of no practice. Several possible explanations were provided to support such a long-term effect [7]. First, the resemblance between the exercise program activities, i.e., the Kata techniques and the stereotypic behavior movements. Second, the model of disordered behavior supported by the optimal stimulation theory [62] and the homeostatic theory [63] that may justify the fact that stereotypy and self-stimulated behaviors decrease after physical-exercise-based interventions. Third, the neurochemical reasons based on abnormal levels of serotonin and dopamine in individuals with autism [64,65], their association with physical exercise and with the maintenance of stereotypic behavior [66,67], particularly in situations of serotonin and dopamine metabolism disturbance as it is the case of hyperseretoninemia [68]. A great deal of further research is needed before these competing explanations can be verified.

The results of this systematic review also provide some useful practical implications. First, physical exercise may contribute to social integration in children with ASD through its effect on reducing from the frequency of stereotypic behavior and its resultant behavior problems. This in turn would contribute to the development of more positive feelings of well-being and quality of life among children with ASD and their families and tutors. Second, physical exercise programs can be used for preventive behavior management and seen as a serious alternative to medication use to inhibit the occurrence of mal-adaptive behavior. Third, and as an indirect practical implication, physical exercise is associated with many health comorbidities in children with ASD, such as those related with higher incidences and higher risk of obesity [69,70,71] and improvements in metabolic health, autistic traits, and quality of life [40].

## 5. Conclusions

This SRM provides evidence for the hypothesis that exercise is effective in reducing the number of episodes of stereotyped behavior in children diagnosed with autism spectrum disorders. Further research is needed to identify which type of exercise intervention programs may be more effective in promoting such positive effects. Given the impact of the stereotypical behaviors’ occurrence in childhood, future research is needed to clarify the impact of PE in children under the age of eight and with other diagnoses, such as Asperger’s syndrome and unspecified developmental disorders, included in the category of the autistic spectrum disorder.

Findings also provide evidence for the need to improve the quality of the design used to plan and implement the physical exercise programs in children with ASD. Future studies should report objective information including frequency, duration, intensity, volume, and type of intervention used as it is important to identify the characteristics and the most appropriate intensity of stimulus necessary to generate effective responses in the reduction of stereotypical behaviors.

Limitations/Future Directions: Although the results of the present systematic review and meta-analysis show a positive effect of exercise it is also important to identify some limitations. First, only a limited number of articles published in the literature analyze the topic of exercise and stereotypical behaviors in pediatric populations with ASD, after using the mentioned inclusion and exclusion criteria. Second, the reduced sample size found in those studies (67 children with ASD) tends to minimize the inferential power over the real effects of physical exercise in this population. Third, there is sample imbalance between genders due to the fact that gender ratio in children with ASD is roughly five males to one female. Fourth, the reduced quality of information provided in many studies, particularly the lack of accurate inferential statistical information and the lack of detailed information about the physical exercise intervention program are quite frequent and were seen as limiting factors for the final selection of manuscripts to be included in the present systematic review with meta-analysis.

## Figures and Tables

**Figure 1 medicina-55-00685-f001:**
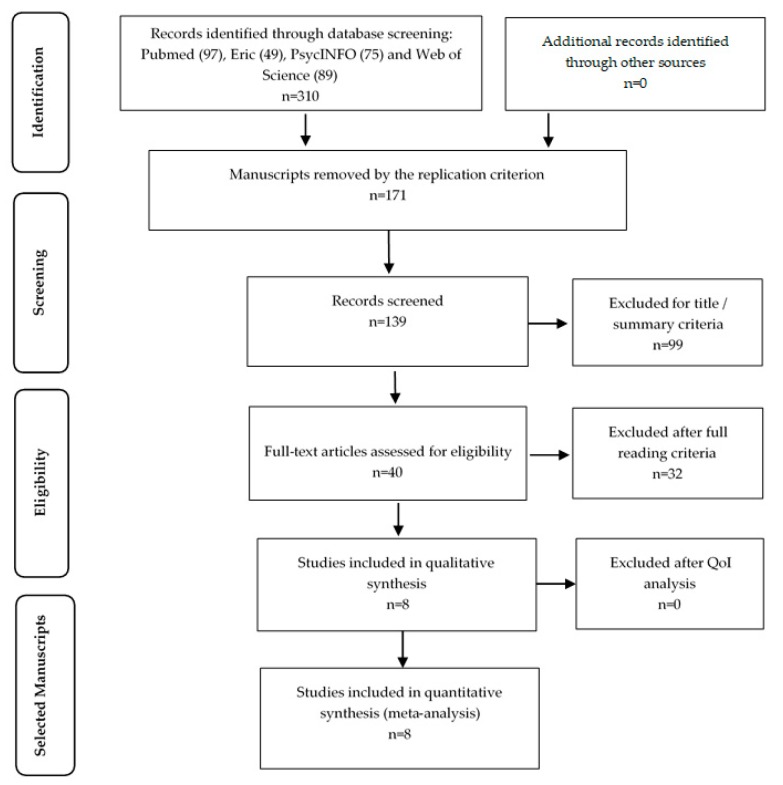
Scheme of information about the different phases of systematic search through the positioning PRISMA guidelines.

**Figure 2 medicina-55-00685-f002:**
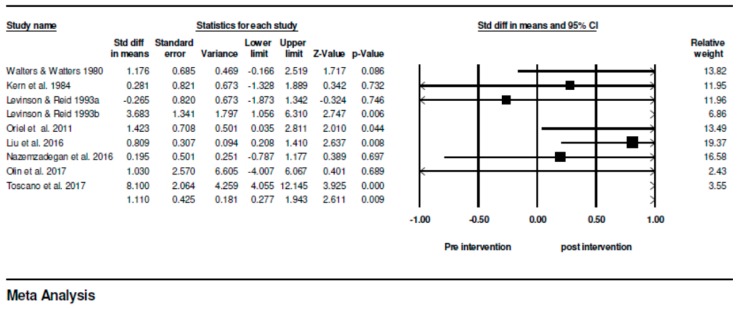
Summary of descriptive and inferential statistics between groups before and after intervention with physical exercise.

**Figure 3 medicina-55-00685-f003:**
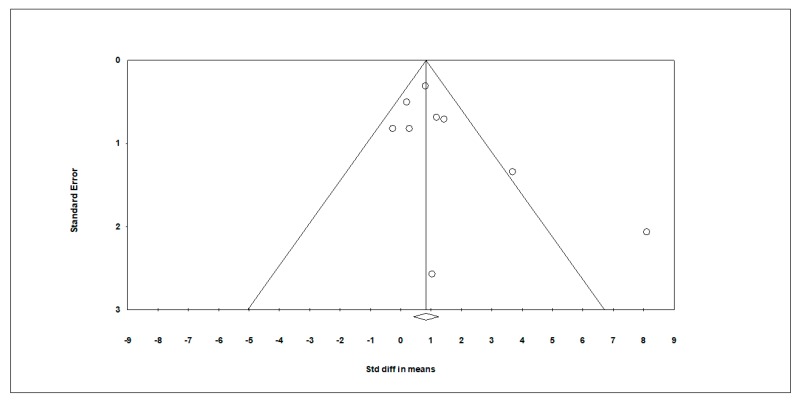
Exercise and occurrence of stereotypic behavior in children with ASD – Funnel plot of standard error by std diff in means.

**Table 1 medicina-55-00685-t001:** Categories and subcategories that emerged from the results in the eight studies selected.

	TREND Assessment Protocol																							
	Title and Abstract	Introduction	Methods/Participants	Methods/Intervention	Objectives	Outcomes	Sample Size	Assignment Method	Blinding (Masking)	Unit of Analysis	Statistical Methods	Results/Participation Flow	Results/Recruitment	Results/Base Line Data	Results/Base Equivalence	Results/Numbers Analyzed	Outcome and Estimation	Ancillary Analyses	Adverse Events	Discussion/Interpretation	Discussion/Generalizability	Discussion/Overall Evidence	Total Items	Total Percentage
Items (paper sections)	I	II	III	IV	V	VI	VII	VIII	IX	X	XI	XII	XIII	XIV	XV	XVI	XVII	XVIII	XIX	XX	XXI	XX	22	100%
Sub-items (descriptor) per items	3	2	4	8	1	3	1	3	1	2	4	7	1	4	1	2	3	1	1	4	1	1	58	100%
1. Kern et al. (1984)																								
Items	1	1	1	1	1	1	0	1	0	1	1	1	1	1	1	1	1	0	0	1	1	1	18	82%
Sub-items	1	2	4	7	1	3	0	3	0	2	1	3	1	3	1	2	1	0	0	4	1	1	41	70.7%
2. Levinson & Reid (1993)																								
Items	1	1	1	1	1	1	0	1	0	1	1	1	1	1	1	1	1	0	0	1	1	1	18	82%
Sub-items	3	2	4	7	1	3	0	3	0	2	1	3	1	3	1	2	1	0	0	4	1	1	43	74.1%
3. Oriel et al. (2011)																								
Items	1	1	1	1	1	1	0	1	0	1	1	1	1	1	1	1	1	0	1	1	1	1	19	86.4%
Sub-items	3	2	3	7	1	3	0	3	0	1	1	4	1	3	1	2	3	0	1	4	1	1	44	75.9%
4. Bahrami et al. (2012)																								
Items	1	1	1	1	1	1	0	1	0	1	1	1	1	1	1	1	1	0	1	1	1	1	19	86.4%
Sub-items	3	2	4	7	1	3	0	2	0	2	2	5	1	4	1	2	1	0	1	4	1	1	47	81.0%
5. Liu et al. (2016)																								
Items	1	1	1	1	1	1	0	1	0	1	1	1	1	1	1	1	1	1	1	1	1	1	20	90.9%
Sub-items	2	2	3	7	1	3	0	2	0	1	3	6	1	4	1	2	3	1	1	4	1	1	49	84.5%
6. Nazemzadegan et al. (2016)																								
Items	1	1	1	1	1	1	0	1	0	0	1	1	1	1	1	1	1	0	0	1	1	1	17	77.3%
Sub-items	2	1	3	6	1	2	0	1	0	0	2	3	1	1	1	2	1	0	0	3	1	1	32	55.2%
7. Olin et al. (2017)																								
Items	1	1	1	1	1	1	0	1	1	1	1	1	0	1	1	1	1	0	1	1	1	1	19	86.4%
Sub-items	3	1	3	6	1	3	0	2	1	1	3	5	0	3	1	2	3	0	1	4	1	1	45	77.6%
8. Toscano et al. (in press)																								
Items	1	1	1	1	1	1	0	1	0	1	1	1	1	1	1	1	1	0	1	1	1	1	19	86.4%
Sub-items	3	1	4	8	1	3	0	2	0	2	4	7	1	4	1	2	3	0	1	4	1	1	53	91.4%

Note: Roman numbers indicate the twenty two items of the TREND assessment protocol. Arabic numbers indicate the number of sub-items fulfilled for each item.

**Table 2 medicina-55-00685-t002:** Synthesis of the systematic search of the scientific literature of the articles selected on the effects of exercise in children with autistic spectrum disorder (ASD).

Author (Year)	Country	Type of Study	Objectives	Sample	Intervention	Data Collection	Data Analysis Method	Outcomes
**1. Kern, Koegel & Dunlap (1984)**	USA	Non-randomized intervention trial	Manipulate two types of physical exercise: vigorous versus less vigorous exercise. Assess whether these activities would influence the children’s subsequent stereotypic and other responding	N = 3 8.33 ± 2.31 years old	A simultaneous-treatments design was used in which sessions of one condition (e.g., 15 min. jogging) were alternated with sessions of the other condition (e.g., 15 min. ball playing).	Scores of stereotypies from direct observation on three conditions: no exercise, with vigorous physical exercise and moderate exercise. Stereotypies observation time 90 min.	Percentage of occurrences differences	Average reduction of stereotypes: (a) Baseline (24.89) (b) Moderate (37.79) (c) Vigorous (22.88)
**2. Levinson & Reid (1993)**	Canada	Non-randomized intervention trial	Examined effects of exercise intensity on the stereotypic behaviors.	N = 3 (2M; 1F) 11.0 ± 0.0 years old	Two treatment conditions: a mild exercise (15 min walking) program and a vigorous (15 min jogging).	Number of occurrence of stereotypic behaviors. Stereotypies observation time 90 min.	Percent agreement ratio differences	Average reduction of stereotypes: (a) pre walking (73 ± 3) post walking (75 ± 8) (b) pre jogg (72 ± 3) post jogg (55 ± 6).
**3. Oriel et al. (2011)**	USA	Within-subjects crossover design Non-randomized intervention trial	Determine whether participation in aerobic exercise before classroom activities improves academic engagement and reduces stereo-typic behaviors.	N = 9 (7M; 2F) 5.2 ± ? years old	The treatment condition included 15 min of running/ jogging followed by a classroom task. The control condition included a classroom task not preceded by exercise.	Number of occurrence of stereotypic behaviors demonstrated, the percentage of on-task behavior, and the numbers of correct/incorrect responses given during academic tasks for each child. Stereotypies observation time 15 + 15 min.	Wilcoxon signed rank test	(a) Intra-class correlation coefficients: correct and incorrect responses, time on task and number of stereotypic behaviors observed (0.97, 0.84, 0.96, and 1.0, respectively). (b) Correct responding: 71.49% on control days vs. 82.57% on treatment days. (c) On-task time and Stereotypic behaviors: no sig. differences.
**4. Bahrami et al. (2012)**	Iran	Non-randomized intervention trial	Assess the effects of 14 weeks of Kata techniques training on stereotypic behaviours of children in autism spectrum disorders.	N = 30 (26M; 4F) 9.13 ± 3.27 years old	Formal exercises Kata technique, 1 session/day, 4 days/week for 14 weeks (56 sessions). The duration of exercise was increased from about 30 min at the start of the program to 90 min after 8 weeks. The time breakdown was as follows: 15 min of warm-up (10 min for stretching, 5 min for jogging), 65 min for the main activity, and 10 min for cool down.	Scores of stereotypies from direct observation by the subscale of Gilliam Autism Rating Scale—2nd edition (GARS-2). Administer stereotypies subscale before and after intervention (30 to 90 min of exercise).	Performed statistical analyses with an independent samples *t*-test, a 2-factor mixed-model ANOVAs, and a paired *t*-test by using SPSS software (Version 11.5).	Average reduction of stereotypes: (a) baseline (12.53 ± 6.92) EG (14.47 ± 7.71) CG (b) post-intervention (7.20 ± 5.65) EG (13.93 ± 8.55) CG (c) follow-up (8.07 ± 5.82) EG (13.40 ± 7.66) CG
**5. Liu et al. (2016)**	USA	Non-randomized intervention trial	Examine the effects of physical activity on stereotypical behaviors of children with ASD.	N = 23 (16M; 7F) 7.6 ± 0.65 years old	Certain activities were offered to the participants assuming that they would raise the child’s heart rate to the desired intensity level such as jumping on a trampoline, stationary biking, completing an obstacle course, and/or playing dance-dance revolution game, MVPA for 15 min.	In a period of 4 h, children were observed for 2 h at baseline and with intervals of 15 min. Then after 15 min of MVPA participation, children were observed for 2 h with intervals of 15 min and recorded as either engaging in stereotypic behavior (SB) or task engaged behavior (TE). Stereotypies observation time 120 min.	Repeated measures ANOVA analysis Effect-size	All children (100%) were engaged in moderate to vigorous intensity level during physical activity participation. For a 4-h observation period, results showed that physical activity reduction effect on stereotypic behaviours in children with ASD lasted for 2 h for. This was determined by comparing the baseline observation data with the post physical activity observation data, which were recorded in the same way using the 15 min’ interval for SB or TE ratings. Significant differences were found on pre and post percentage scores, F (1, 17) = 7.523, *p* = 0.009 < 0.05. No significant difference was found on gender, F (1, 17) = 4.253 *p* = 0.009 > 0.05, and disorder F (2, 17) = 2.949, *p* = 0.009 > 0.05.
**6.Nazemzadegan et al. (2016)**	Iran	Non-randomized intervention trial	Evaluating the effectiveness of ball exercises on the reduction of stereotypical behavior of children with autism spectrum disorder	N = 8; 9.0 ± 2.5 years old	Quasi-experimental with pretest-post-test and control group. Sixteen children were selected and were randomly assigned to two groups with eight children each (test and control). Both the experimental and control groups were placed under the Jim ball intervention training for 24 sessions (12 weeks, 2 sessions of 45 min of exercise). Individual training sessions were conducted for each subject.	Scores of stereotypies from direct observation by the subscale of Gilliam Autism Rating Scale—2nd edition (GARS-2). Administer stereotypies subscale before and after intervention (45 min of exercise).	Kolmogorov-Smirnov Test Regression analysis of covariance	Analysis of variance showed a significant difference in post-test scores of both the experimental and control groups (*p* = 0.01), thus showing the effectiveness of the intervention. Results suggested that the Jim exercise ball could change the stereotypical behavior of children with high functioning autism spectrum disorder.
**7. Olin et al. (2017)**	USA	Non-randomized intervention trial	Quantify the acute effect of exercise and to assess the influence of duration and intensity on the frequency of stereotypic behaviors in children with ASD	N = 7; 13.0 ± 1.4 years old	Participants were randomly assigned to exercise or control group. Exercisers self-selected a stationary bike, a treadmill or an elliptical ergometer. Continuous aerobic exercise was performed at a low (L) or high (H) intensity for either 10 or 20 min. Four exercise sessions served as experimental conditions: 10L,10H, 20L, and 20H. Intensity was: L between 50% and 65% age-HRmax and between 70% and 85% HRmax via continuous measurement, along with assessment of a rating of perceived exertion every 3–5 min using the OMNI scale.	Participants exhibited observable forms of self-stimulation and other inappropriate behaviors. As these behaviors are different from person to person, each subject´s mannerisms were determined and used for the subsequent measure. The primary SSB identified were hand flapping and echolalia. Stereotypies observation time 60 min.	Repeated-measures 5 × 5 (condition × time) ANOVA, and interaction effects followed-up by examining simple effects of condition within time.	A sig. condition main effect (*p* < 0.01) and a sig. time condition interaction effect (*p* < 0.10) were found. Sig. diff. between conditions in the first 15 min after exercise (*p* < 0.05). Pairwise comparison indicated that 20H produced sig. worse SSB than all other conditions (*p* < 0.10) except control (*p* < 0.10). Sig. diff. between conditions at post30 (*p* < 0.05), with 10L with greater decrease in SSB than 20L or 20H (*p* < 0.10). Control differed only from 20H (*p* < 0.10). Sig. diff. between conditions at post45 (*p* < 0.08) due to the fact that 10L produced sig. greater reductions in SSB compared with all other conditions (*p* < 0.10). Diff. between conditions also at post60 (*p* < 0.01), with 10L producing better effects on SSB than Control or 20H (*p* < 0.05).
**8. Toscano et al. (in press)**	Brazil	Randomized intervention trial	Effects of a 48-week exercise-based intervention on the metabolic profile, autism traits and perceived quality of life in children with ASD	N = 46; 8.2 ± 1.7 years old	Climbing and support in the bar (Upper limb strength); Release to the basket (Upper limb strength); Work with elastics (Strength of the lower and upper limbs); Walking on steps and inclined plane and step box with target and sequenced march (Strength, Coordination).	Autistic traits scale values decreased across the 48-week of intervention. Administer stereotypies subscale before and after intervention (30 min of exercise for 48 weeks).	Multilevel modeling to examine the responses to the 48-week exercise-based intervention program	Participants from the intervention group revealed a decrease across the 48-weeks of intervention with physical exercise (−8.1, 95% CI, −12.2 to −4.0, effect size = 1.05).

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
