# Peer review of "Effects of Physical Exercise on the Stereotyped Behavior of Children with Autism Spectrum Disorders"

_medicina, 2019, doi:10.3390/medicina55100685_

Round 1

Reviewer 1 Report

A systematic review and meta-analysis is an important step in the identification of appropriate interventions for children with autism spectrum disorder (ASD).  The present manuscript attempts to identify the potential effectiveness of prescribed physical exercise on the stereotypical restrictive and repetitive behaviors in children with ASD.  This is a worthwhile and timely endeavor.  However, the paper in its current form lacks consistency in the analysis of the studies selected and forms a conclusion on a select few of the papers reviewed.  The paper is also difficult to read in its current grammatical form and contains sections that while substantial in statistical analyses, do not add to the overall value, nor are addressed in the discussion and conclusion.  The references are incorrectly formatted and the format changes throughout the paper from numeric to author names and back again. (e.g., Line 76-77).  The format for effect size is the italicized (d) throughout the paper. This lack of polish on the paper indicates a potential concern for lack of attention to detail on other sections. Though a full and detailed review of each error should be presented, at this time the overarching issues in the methodology and presentation need to be addressed prior to the line-by-line minor grammatical and format review. I have tried my best to comment of several sections needing improvement.

Methodologically, the paper fails to identify how long each observation session for the evaluation of behavior lasted.  What was the sampling rate of the observations made in each study? (e.g., 1 every 10 seconds for 20 minutes?, number of occurrences over 15 minutes) How can these differences be addressed? The design in section 2.4 is redundant and overly descriptive of the PRISMA and PICOS formats

The paper switches between reviewing the 8 selected paper and then refers to 9 entries.  This was confusing and confounds the description of the conclusions reached.  The Data extraction section is confusing to read and includes generalities that need to be further described, (e.g., is the third and independent reviewer truly independent or are they a co-author? What happens to an article classified as "doubtful" versus included or excluded?)

The criteria for study selection (Line 118) should be identified prior to a discussion of data extraction for clarity and the number of stereotyped behavior episodes need a time frame reference. (Line 126-7)

Addressing publication bias, while important should not be a focal point of the work.  Section 2.6 concludes with a paragraph needing substantial organizational work and additional clarity.  This analysis is not common. While section 2.8 attempts to describe the process used for the identification of heterogeneity of findings, the sheer number of analyses included seems to be a "throw everything at it and see what sticks" approach.  Selectivity and justification in analytic tools is an important consideration to make and is absent in this paper.

In the Results sections, Figure 1 is confusing as the second block from the top is an exclusion block while the other exclusion blocks are to the right of the main column.  There is also a failure to identify why studies were excluded at each stage in the figure.  The narrative then goes on to repeat the representation in the text.  The authors should chose one or the other and not both. Table 1 is not traditionally intuitive and the presence and absence of a variety of horizontal lines clouds the information further.  Why is one row Roman numeral and others Arabic?  Does a "1" indicate the presence of the quality?  If so, what does a 2 or 4 indicate.  A well-developed legend would help.

Section 3.2 attempts to describe the participants in the selected studies but fails to identify some key components. Line 221 addressed "high reported stereotyped behaviors". Who is reporting this for each? What are the diagnostic characteristics of the children?  A well known limitation of studies in this population is the heterogeneity of the ASD presentation.  The current paper fails to his in the participant selection and also fails to address this in the limitations.  This could confound the data on which the conclusions are justified on.  In Table 2 - trial is misspelled in each instance and there is not measurement frequency presented.  Further EUA requires a definition.

The outcomes listed in section 3.3 are not true outcomes of the meta-analytic process, they seem to be a mix of methodology and narrative descriptions of the hypotheses for the selected studies.  Lines 265-6 describe ambiguous statements such as "in the universe of studies".  Truly the authors do not mean "the universe"?  That in and of itself is a bit presumptuous. Second a description of the meaning of confidence intervals is unnecessary ads the reader should have this ability to interpret the results.  Yet, be sure to clarify that in Table 3 the lower and upper limits presented are indeed 95% confidence intervals.  This is currently unclear. It is also unclear why there are 10 lines for 8 studies or 9 entries.  Additionally, please indicate that the last line of data is a sum of the parts above it.

I recommend excluding or significantly modifying the section 3.5 on "Homogeneity of the effects".  Prior section refer to the heterogeneity of effects but it is the opposite here.  Additionally, I will state that this isn't a standard interpretation and the large number of statistical variables presented does nothing more that confuse the interpretation of results. The paragraph from Line 296-306 describes a plot narrowing as the sample size increases, yet according to Figure two, this occurs as the Standard Error decreases.  Additionally the intercept described as ~1.515 on Line 302 does not appear in the figure where the x-axis intercept seems to be under 1.0.  

In Section 4 there are significant portions in need of revision and consideration including clarification of numerous generalities (e.g., Line 317 - 8 studies vs. 9 entries, Line 333 - "type of exercise" (mode?), Line 320 - "repertoire", Line 339-41 - effort = intensity whereas aerobic = metabolic process, Line 351 - "relative weight", Line 356 "MVPA" etc.) 

With these substantial errors and omissions I am unable to recommend this paper for publication.

Reviewer 2 Report

The authors conducted MA and SR of the influence of physical activity interventions on stereotypical behavior of children with ASD. The authors concluded that regular physical activity appear to have the potential to positively impact on stereotypical behavior of children with ASD.

The study idea is very important; the MA and SR have been conducted following the rules of performing MAs and SRs. 

Abstract: please add mean age and gender ratio; report, if data of children with ASD were compared to healthy controls, or ‘simply’ as pre-post comparisons. Further, explain briefly, why outcome variables were stereotyped behavior and not (also) social interactions.

Introduction: Please report the key features of ASD, gender ratios and explain, as to why prevalence rates might differ between continents. Specifically, describe in more details that persons with ASD suffer from social deficits and from the inability to perceive social cues and to interact adequately.

Describe to whom ‘Portuguese mainland population’ does refer to, and it is important to mention the prevalence rate of this population.

The authors mention a range of treatments (e.g., antecedent-based treatments), though, the authors should explain in more details, if these could be considered psychotherapeutic interventions, and if so, the authors should give some results/ information of effectiveness. 

L 52; define in more details ‘episodes’, as this outcome variable is the key to understand the entire pattern of results.

L 76; please revise reference style.

L 82; should be autism

The methodological procedure was particularly well described.

Results: again, underscore the comparators: which whom were the results of children with ASD compared to?

Discussion: please carefully check the reference style, e.g. L 388.
